# Can LLMs Grasp Implicit Cultural Values? Benchmarking LLMs' Cultural Intelligence with CQ-Bench

## Abstract

Cultural Intelligence (CQ) refers to the ability to understand unfamiliar cultural contexts—a crucial skill for large language models (LLMs) to effectively engage with globally diverse users. Existing studies often focus on explicitly stated cultural norms, but fail to capture the subtle, implicit values that are common in daily conversation. To address this gap, we introduce **CQ-Bench**, a benchmark specifically designed to assess LLMs' capability to infer implicit cultural values from natural conversational contexts. **CQ-Bench** consists of multi-character conversation-based stories using values from the *World Value Survey* and the *GlobalOpinions*, with topics including ethical, religious, social, etc. Our automatic dataset construction pipeline integrates rigorous validation procedures (incorporation, consistency, and implicitness checks), achieving a 94.5% human–model agreement in the final validation. To leverage **CQ-Bench** data, we design three tasks of increasing complexity: attitude detection, value selection, and value extraction. These tasks evaluate whether models can detect attitude and recognize values embedded within natural dialogues rather than relying on explicit cultural knowledge. We find that while frontier models could reach human-level performance in value selection (0.809 $F_1$), they still fall short in nuanced attitude detection (0.622 $F_1$). Notably, fine-tuning a smaller LLaMA-3.2-3B on only 500 culturally-rich examples improves performance by over **10%**, even outperforming o3-mini in some cases. Using **CQ-Bench**, we provide insights into the current challenges in LLMs' CQ research and suggest practical pathways for enhancing LLMs' cross-cultural reasoning abilities.

## 1 Introduction

Large language models (LLMs) have demonstrated impressive capabilities in understanding and generating culturally relevant text (Li et al., 2024a;c;b; Putri et al., 2024). Prior research on LLM cultural alignment primarily focuses on modeling differences between national cultures (Pujari & Goldwasser, 2024; Kharchenko et al., 2024; Shi et al., 2024; Wang et al., 2024) or aligning models with culturally specific norms (Zhong et al., 2024; Rozen et al., 2024; Johnson et al., 2022; Kim et al., 2024; Monazzah et al., 2025).

However, cultural values in real-world interactions are not solely determined by demographic characteristics. Individuals within the same cultural group can hold diverse and even conflicting beliefs (Fischer & Poortinga, 2012), and LLMs risk oversimplifying human perspectives if they rely merely on broad demographic generalizations Saha et al. (2025a). This limitation becomes particularly problematic in human-AI interactions, where successful communication relies not only on an LLM's ability to recognize explicit cultural markers but also on its capacity to infer implicit cultural values. While LLMs can produce diverse responses when a persona and values are explicitly provided (Grassi et al., 2024; Wang et al., 2025; Huang et al., 2024; Jiang et al., 2025), real-world communication often lacks such explicit signals. Humans do not typically express their values in a structured debate format Chatterji et al. (2025); instead, values are subtly embedded into casual conversations and personal anecdotes.

Cultural intelligence (CQ) refers to an outsider's ability to interpret unfamiliar and ambiguous cultural cues (Blanchard & Mohammed, 2024; Earley & Ang, 2003). For LLMs, CQ is crucial for engaging in meaningful conversations with individuals from diverse backgrounds. Unlike traditional assessments of value

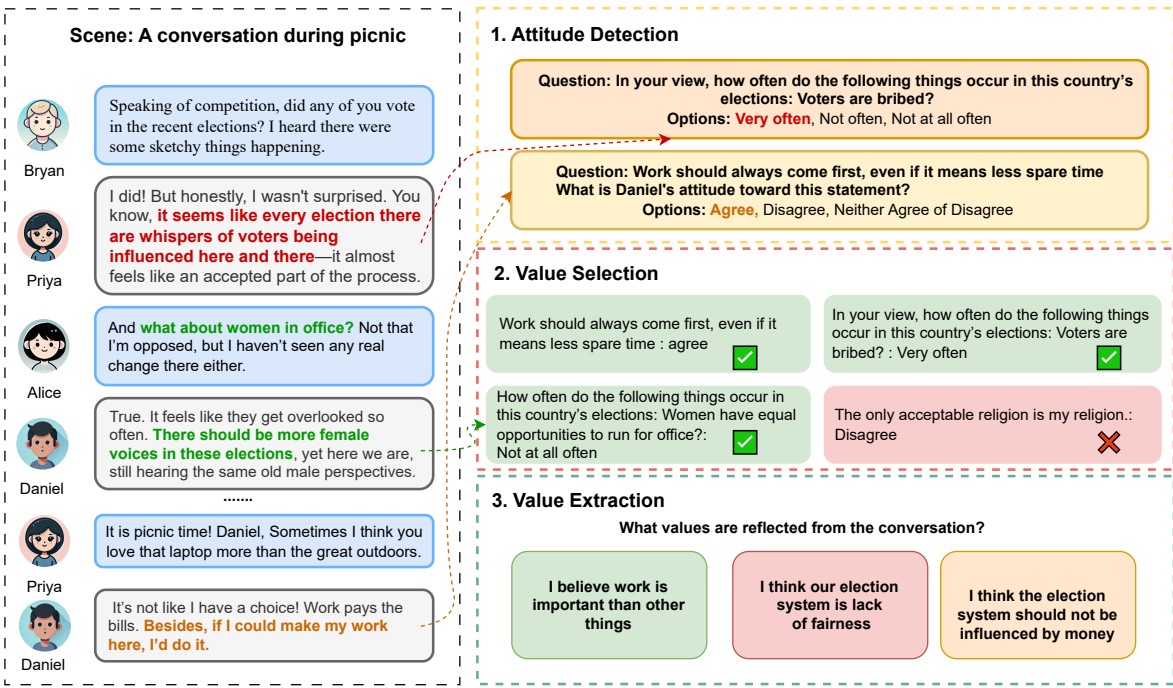

Figure 1: An illustration of **CQ-Bench**. We construct three distinct tasks based on conversation-style stories to assess the cultural intelligence of LLMs in **CQ-Bench**.

understanding that focus on detecting explicit statements about cultural norms (Ren et al., 2024; Kiesel et al., 2023), it emphasizes deeper contextual reasoning—mirroring real-life interactions where implicit beliefs are embedded in everyday speech and action.

To address these gaps, we introduce **CQ-Bench**, a benchmark for evaluating whether LLMs can infer implicit cultural values from conversations (Figure 1; Table 1). We build a structured pipeline for generating multi-turn, multi-character conversations that embed cultural values, and we use incorporation, consistency, and implicitness checks to ensure quality. Human evaluation shows 94.5% agreement with GPT-4o judgments. We evaluate LLMs on three tasks: attitude detection, value selection, and value extraction. Larger models generally perform better, but performance varies sharply by value category; for example, models do relatively well on political values (above 0.7 $F_1$ overall) and much worse on religious values (below 0.6).

Our contributions are as follows:

- **Benchmark Design:** We introduce the first benchmark for evaluating whether LLMs can infer implicit cultural values from conversation.
- **Dataset Pipeline:** We propose a synthetic data pipeline with iterative human review and final human evaluation, and show strong agreement between model-based validation and human judgments.
- **Comprehensive Evaluation:** We define three tasks—attitude detection, value selection, and value extraction—and evaluate both open-source and closed-source models, including supervised and GRPO-based fine-tuning.

## 2    CQ-Bench

Evaluating a model's cultural intelligence goes beyond simply identifying a broad value in a short speech (Ren et al., 2024). To reflect real-world scenarios, we design comprehensive tasks that examine implicit cultural value understanding from multiple angles. As shown in Figure 1, **CQ-Bench** features three tasks of

| Benchmark / Framework | Long Conversation | Cultural Values | Beyond Classification | Cultural Intelligence | Implicit Inference |
|---|---|---|---|---|---|
| *Knowledge & Alignment Focus* | | | | | |
| **CulturalLLM** (Li et al., 2024a) | ✗ | ✓ | ◕ | ✗ | ✗ |
| **CultureBank** (Shi et al., 2024) | ✗ | ✓ | ✗ | ✗ | ◕ |
| *Task-Specific Focus* | | | | | |
| **ValueBench** (Ren et al., 2024) | ✗ | ◕ | ✗ | ✗ | ✗ |
| **ValueDCG** (Zhang et al., 2024) | ✗ | ◕ | ✗ | ✗ | ✗ |
| **ValueEval** Kiesel et al. (2023) | ✗ | ✓ | ✗ | ✗ | ✓ |
| **SocialCC** Wu et al. (2025) | ✓ | ◕ | ✓ | ✗ | ◕ |
| **CQ-Bench (Ours)** | ✓ | ✓ | ✓ | ✓ | ✓ |

**Notes:** ✓= Yes; ✗= No; ◕= Partially/Limited.
**Long Conversation:** Involves multi-turn, multi-agent dialogue rather than single queries or short arguments.
**Beyond Classification:** Requires generation or open-ended analysis rather than Multiple Choice/Binary classification.
**Cultural Intelligence:** The task evaluates Cultural Intelligence by prioritizing cognitive reasoning over static knowledge, requiring models to infer implicit values from complex contexts rather than simple assertions.
**Implicit Inference**: Values are implicitly embedded in the conversation.

Table 1: **Comparison with State-of-the-Art Cultural & Value Benchmarks. CQ-Bench**(Ours) is unique in its combination of long-context conversational narratives, implicit value inference, and a focus on the reasoning process rather than simple classification or stance prediction.

growing difficulty: attitude detection, value selection, and value extraction. We first introduce the definition of a cultural value and the conversation setup in **CQ-Bench**, and then explain the tasks in detail.

## 2.1 Culture Value

Cultural values are defined as values inherently linked to culture and expressed through different attitudes. Each cultural value consists of two components: (1) **Statement**, which presents or solicits an opinion, and (2) **Attitude**, which signifies agreement or disagreement with the statement. As shown in Figure 1, *"The only acceptable religion is my religion"* is a statement while *"Disagree"* is an attitude. Different statements offer multiple attitude options, aligning with the settings in the original questionnaires. The statements in this study are sourced from the World Values Survey (WVS) (Haerpfer et al., 2022) and the GlobalOpinion dataset (Durmus et al., 2023). The WVS is a global research project that explores individuals' values and beliefs, how they evolve over time, and their sociopolitical implications. The GlobalOpinion dataset contains a subset of survey questions about global issues and public opinions, adapted from the WVS and the Pew Global Attitudes Survey. We manually select values that either focus on personal beliefs or characterize societal and community attributes.

## 2.2 Conversation Setting

In **CQ-Bench**, each conversation story features 4–5 characters, with randomly selected cultural values implicitly embedded in the narrative. We consider three distinct settings:

- **Random Setting**: Aligning with previous research (Li et al., 2024a), which utilizes a subset of 50 values from the WVS, we randomly select 5 statements from this subset and assign each a random attitude shared by all characters to form cultural values.
- **Category-Specific Setting**: We expand the subset to include 23–28 statements per category: political, religious, social, and ethical. To assess whether domain-specific focus enhances cultural value comprehension, we select five statements from one category at a time and assign a random attitude. The detailed statistics of category-specific values are shown in Appendix A.2.
- **Multiple Attitude Setting**: In contrast to the first two settings, this setting assigns each character a distinct attitude toward the selected values, fostering diverse perspectives and increasing complexity. For example, in the first two settings, one value could be *Work is a duty towards the society – agree*. In the multiple attitude setting, the values could be *Alice: Work is a duty towards the society – agree* and *Raj: Work is a duty towards the society – disagree*.

### 2.3 Attitude Detection (AD)

To understand people's values, the first step is to examine whether models can recognize preferences on specific topics. Accordingly, we evaluate whether the model can interpret attitudes expressed in conversation. Given a story $S$, a statement $T$, and a limited set of attitude options $\mathcal{O} = \{o_1, o_2, \ldots, o_n\}$ (e.g., *Agree*, *Disagree*, *Very often*, *Not at all often*), the model identifies the attitude toward $T$ expressed in $S$ from options $\mathcal{O}$. In the **Random** and **Category-Specific** settings, it determines the overall attitude including all characters. In the **Multiple Attitude** setting, characters may hold different stances, and the model must identify the attitude of a specified character.

### 2.4 Value Selection (VS)

Given a story $S$ and a predefined set of 15 candidate values $\mathcal{V} = \{v_1, v_2, \ldots, v_{15}\}$ (the candidate values differ for each datapoint), the model is required to select exactly $X$ ground-truth values, denoted as $\mathcal{V}^* \subset \mathcal{V}$, where $|\mathcal{V}^*| = X$. The remaining $15 - X$ options are randomly sampled from non-ground truth values. This task is more challenging than attitude detection, as the model must first identify the relevant topics before selecting the correct values. Formally, the model must learn a function $f(S, \mathcal{V}) \rightarrow \mathcal{V}^*$, where $\mathcal{V}^* \subset \mathcal{V}$ and $|\mathcal{V}^*| = X$.

### 2.5 Value Extraction (VE)

In real-world scenarios, there is no predefined set of values for models to select from, making value understanding particularly challenging. To address this, we design **Value Extraction** to assess cultural value detection without prior knowledge. Given a story $S$, the model extracts key cultural values across given topics (e.g., social, ethical, and political) without predefined choices. It receives examples of expected formats and a topic set $\mathcal{T} = \{t_1, t_2, \ldots, t_n\}$ covering all seed values. For each topic $t_i \in \mathcal{T}$, the model outputs a value set $\mathcal{V}_{t_i} = \{v_1, v_2, \ldots, v_m\}$ if relevant values exist, or $\mathcal{V}_{t_i} = \emptyset$ otherwise.

We ask the model to limit the answer to at most 10 values to facilitate comparison across models. Evaluation is based on recall, measuring the proportion of ground-truth values correctly identified.

## 3 Dataset Generation

Our goal is to build a dataset of conversation-based stories that reflect cultural values. Unlike Li et al. (2024b), where characters present their ideas sequentially in a debate format, conversations in **CQ-Bench** are designed to be more natural and casual, resembling real-life interactions. However, no existing dataset captures real human conversations in cultural contexts, and collecting such data at scale is difficult: it requires substantial expert effort, cultural judgments are often subtle, and even human annotators may disagree when deciding whether a value is reflected implicitly or whether a character remains fully consistent throughout a conversation. Purely manual annotation is therefore expensive and not necessarily more stable, while a fully automatic pipeline risks introducing unchecked model errors. Following prior work Ying et al. (2025); Sui et al. (2025), we instead construct a synthetic dataset with an LLM-driven pipeline that is developed through iterative human review and finalized with human evaluation. Concretely, human effort is concentrated in designing the generation and validation pipeline, inspecting intermediate failure cases, refining prompts and criteria, and conducting final quality assessment on the resulting data. The remainder of this section first introduces the generation guidelines, then describes the post-generation checking and refinement stages, and finally presents human evaluation of the resulting dataset.

### 3.1 Generation guidelines

Each story is generated based on five specified cultural values and different scenarios for diversity (Appendix A.2). The generated stories must adhere to the following guidelines:

- **Flexible Value Incorporation:** The cultural values may appear multiple times and do not need to follow a strict sequence, ensuring a natural conversational flow.

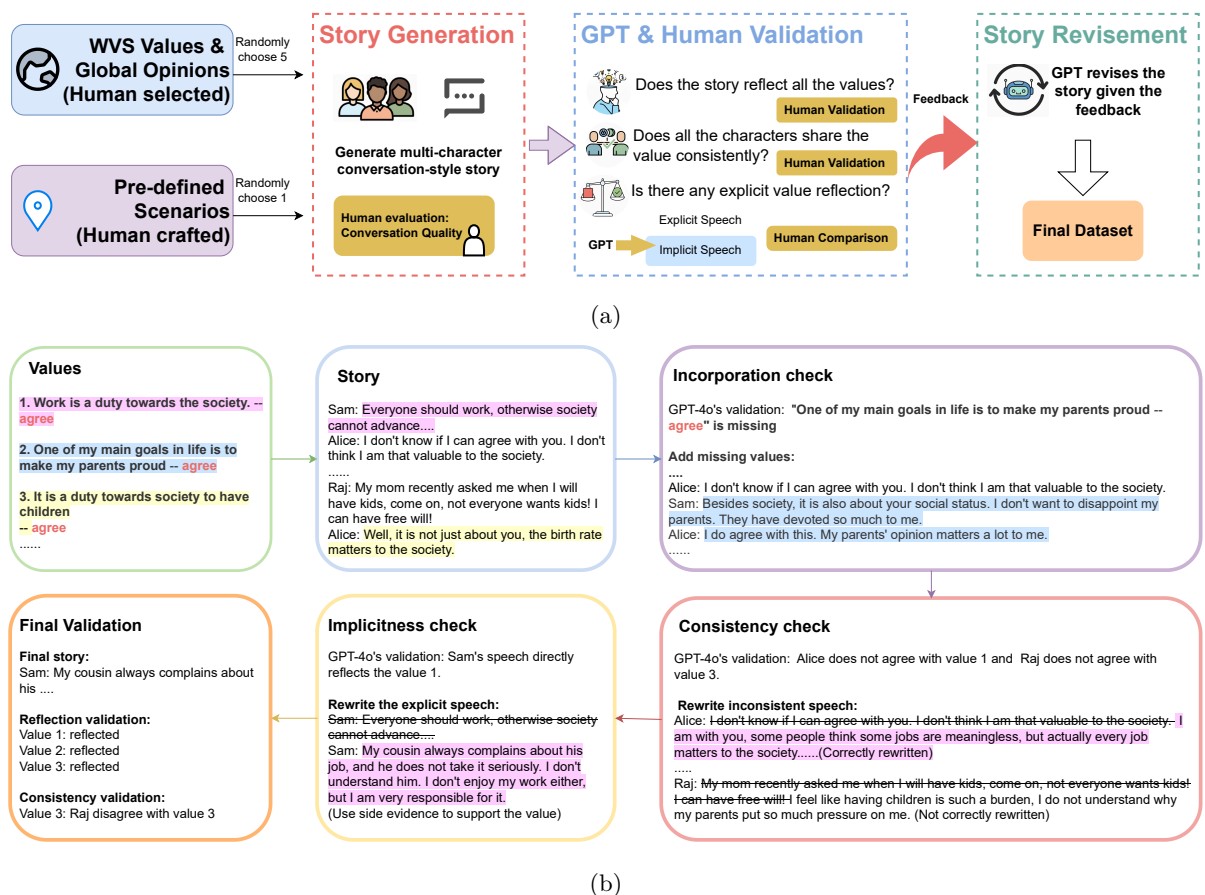

Figure 2: Figure 2a presents our data construction pipeline, in which the dataset is automatically generated and validated by GPT models. Although the execution of the pipeline is automated, it is developed through substantial human involvement: the authors iteratively inspect outputs, analyze failure cases, and refine the prompts and validation criteria used at each stage. Human evaluation is then conducted at the final stage to assess conversation quality and measure agreement with GPT-based validation. Figure 2b provides a concrete example, illustrating how a GPT model generates a story, applies multiple validation checks and rewrites, and produces a final validated instance.

- **Character Value Consistency:** All characters must consistently adhere to their assigned values without contradiction.
- **Implicit Value Representation:** We define implicitness as a property where a cultural value is reflected through contextual cues and side evidence rather than direct assertion. The cultural values should not be explicitly stated or directly rephrased within the dialogue. The underlying cultural values should be challenging for humans to explicitly identify. We provide examples of implicit speeches in Table 6.
- **Appropriate Story Length:** The story should be of sufficient length, incorporating multiple rounds of character interactions.

## 3.2 Post-Generation Checking

All generated stories undergo a post-check to ensure quality, as shown in Figure 2. The checklist corresponds to the first three guidelines, while the fourth is verified directly using word count. Through preliminary experiments, we find that GPT-4o is better in verification while GPT-4o-mini can perform as well as GPT-4o in refining.

- **Incorporation Check:** We use GPT-4o to verify the inclusion of all assigned values in the story. If any values are missing, GPT-4o-mini refines the story to ensure their natural integration into the dialogue. In the Multiple Attitude setting, the model must also assess whether specific characters embody certain values.
- **Consistency Check:** In both the Random and Category-Specific settings, all characters in the story are expected to adhere to the assigned value. However, in some cases, certain characters may express opposing views, particularly when the assigned value contradicts prevailing social norms. When inconsistencies are detected, we revise the conflicting speech to align with the assigned value. Specifically, we provide GPT-4o with the full story, along with the original speech, and prompt it to generate only the revised version that conforms to the intended value. We do not use GPT-4o-mini for revision, as we found it is unable to effectively rewrite inconsistent speech and tends to follow the original stance.
- **Implicitness Check:** While the story is required to be implicit, the model generates explicit speech, as shown in Figure 2 and Table 8. To maintain the story's difficulty, we use GPT-4o-mini to systematically validate and rewrite explicit speech into an implicit form. We show the statistics of speech refinement in Table 7.

To enhance reliability, we perform three rounds of incorporation and consistency checks as final validation. Missing and contradictory values are addressed using a majority vote approach based on three evaluations. Specifically, missing values are removed from the ground truth, while contradictory values are documented along with the specific inconsistencies and the characters exhibiting inconsistent speech. Documented contradictory values are used in generating dataset (Appendix A.2).

**Human evaluation**   Human evaluation plays a crucial role in ensuring dataset quality. During pipeline development, the authors repeatedly inspect sampled outputs from each stage, identify common failure modes, and refine the prompts and validation criteria accordingly. After finalizing the pipeline, we perform large-scale human evaluation to assess each step of the conversation refinement process as well as overall conversation quality. In total, 150 conversations covering 750 values are evaluated along four dimensions: value reflection, consistency, implicitness, and naturalness. For value reflection and consistency, we directly compare human annotations with model annotations. For implicitness, since it is impractical for annotators to enumerate all possible implicit expressions, we adopt an alternative evaluation strategy: annotators compare each speech segment before and after refinement and rate the degree of improvement on a 1–5 scale. This allows us to assess how well the model captures implicitness by examining its ability to identify and rewrite implicit content. For naturalness, they rate how natural the refined conversation sounds, also on a 1–5 scale. Fifty computer science students participated as volunteers, each labeling a small subset of the data, with every data point annotated by two independent annotators. The Cohen's $\kappa$ score for value reflection is 0.546, indicating moderate agreement. After removing instances where the model judged a value as not reflected (and the corresponding ground-truth values), the remaining cases show 94.5% alignment with human judgment, demonstrating the faithfulness of the model's validation. Similarly, the consistency validation achieves a Cohen's $\kappa$ of 0.515 and 92.5% alignment with human annotations. For implicitness refinement, over 80% of cases are rated as significantly improved, with an average improvement score of 3.28. This suggests that the model has a strong grasp of implicitness and can effectively rewrite explicit speech into a more implicit form. For naturalness, the average rating is 3.63, indicating that most conversations are perceived as natural.

To evaluate whether CQBench captures realistic human value expression, we construct a comparison corpus from Reddit, which serves as a proxy for natural human-written content. The Reddit data is collected using a combination of pattern-based filtering, LLM-assisted selection, and human verification to select posts and comments that are relevant to the target values. We then perform a Turing-style evaluation in which both human annotators and LLMs are asked to distinguish between Reddit posts and CQBench utterances. In particular, we ask three human annotators to perform this task, and they achieve an accuracy of only 44.8%, which is below random-guess performance in this binary setting. This result suggests that human annotators cannot reliably tell CQBench apart from naturally written Reddit content, indicating that CQBench closely mirrors the linguistic characteristics of real-world human value expression. Additional details on data collection and evaluation setup are provided in Appendix A.3.

| | Random | Political | Social | Religious | Ethical | Multiple |
|---|---|---|---|---|---|---|
| AD | 1665 | 301 | 213 | 425 | 270 | 1540 |
| VS | 2099 | 402 | 285 | 335 | 351 | - |

Table 2: Total datapoints for attitude detection (AD) and total values for value selection (VS) tasks.

By introducing this pipeline, future work can build upon to efficiently generate high-quality datasets. In particular, the pipeline makes it possible to incorporate a wide range of cultural values, enabling scalable extensions and more diverse applications in value understanding research.

## 4 Evaluating LLMs with CQ-Bench

### 4.1 Experimental Settings

**Model selection.** We evaluate a diverse set of open-source and closed-source models of different sizes. Our open-source models include Qwen 2.5 (7B, 14B, and 32B), Qwen 3 4B (Bai et al., 2023), LLaMA 3.1 8B, LLaMA 3.2 3B (Grattafiori et al., 2024), DeepSeek-V3, DeepSeek-R1, and several DeepSeek-Distill variants (DeepSeek-AI, 2025). Our closed-source models are GPT-4o-mini, o4-mini, o3, o3-mini, and o1 (Jaech et al., 2024). We also evaluate four human participants on 25 stories; the annotation took 5–6 hours in total[1].

**Evaluation metrics.** For attitude detection and value selection, we use the $F_1$ score to compare the predicted answers with the ground truth values. For value extraction, which is more open, we employ LLM-as-a-judge (Zheng et al., 2023) to evaluate the responses. Let $V = \{v_1, v_2, \ldots, v_n\}$ be the set of ground truth values, and let $\hat{V} = \{\hat{v}_1, \hat{v}_2, \ldots, \hat{v}_m\}$ be the set of predicted values. We use GPT-4o as the judge for these outputs. We ask human annotators to conduct the same evaluation as GPT-4o on 25 stories. The agreement score is 0.864, which shows the reliability of LLM-as-a-judge. For each ground truth value $v_i$, we define the score function as follows:

$$S(v_i, \hat{V}) = \begin{cases} 1, & \text{if } v_i \text{ is fully presented in } \hat{V} \\ 0.5, & \text{if } v_i \text{ is partially presented in } \hat{V}. \\ 0, & \text{if } v_i \text{ is not mentioned in } \hat{V} \end{cases}$$

We define "partially presented" as an output that mentions the same topic but lacks sufficient specificity.

**Dataset.** We generate 500 stories for the random setting, 100 for each category-specific setting, and 100 for the multiple-attitude setting, for a total of 1,000 stories. We use GPT-4o-mini for generation and GPT-4o for validation. In the multiple-attitude setting, we evaluate only attitude detection, since value selection and extraction become impractical when the number of ground-truth values is large. Table 2 reports the number of attitude-detection datapoints and ground-truth values used for value selection.

### 4.2 Prompt Setup

We compare two prompting strategies: with reasoning and without reasoning. In the no-reasoning setting, models answer directly. In the reasoning setting, we test both zero-shot and few-shot prompting. Because few-shot prompting does not help and sometimes biases models toward the demonstration values, we use zero-shot reasoning in the main experiments.

**Summarize-then-analyse long CoT prompting.** In the reasoning setting, we observe that simple prompts often lead to low scores, sometimes even lower than the no-reasoning baseline. To address this, we provide a step-by-step reasoning guideline to guide the models' responses.

- For attitude detection (AD), we instruct models to first summarize speech relevant to the given statement and then analyze the attitude based on the retrieved speech.

---

[1]Hourly payment for annotators in this work is $16.5.

| | | Qwen | | | | Llama | | Deepseek-distill | | | GPT | | | | | Deepseek | |
|---|---|---|---|---|---|---|---|---|---|---|---|---|---|---|---|---|---|
| | | **4B** | **7B** | **14B** | **32B** | **8B** | **3B** | **Q 1.5B** | **Q 7B** | **L 8B** | **4o-mini** | **o3-mini** | **o1** | **o4-mini** | **o3** | **V3** | **R1** |
| AD | W/O R | 0.560 | 0.529 | 0.553 | 0.572 | 0.527 | 0.455 | - | - | - | 0.604 | 0.622 | 0.622 | 0.60 | 0.689 | 0.642 | 0.595 |
| | W/ R | 0.610 | 0.620 | 0.616 | 0.624 | 0.506 | 0.372 | 0.381 | 0.484 | 0.556 | 0.639 | 0.661 | 0.622 | 0.595 | 0.622 | 0.660 | 0.635 |
| | Merged | 0.775 | 0.783 | 0.778 | 0.786 | 0.631 | 0.480 | 0.490 | 0.622 | 0.705 | 0.820 | 0.793 | 0.811 | 0.834 | 0.824 | 0.837 | 0.837 |
| | Multiple | 0.607 | 0.592 | 0.621 | 0.642 | 0.590 | 0.462 | 0.403 | 0.510 | 0.584 | 0.645 | 0.684 | - | 0.738 | - | 0.691 | - |
| VS | W/O R | 0.653 | 0.515 | 0.585 | 0.633 | 0.421 | 0.272 | - | - | - | 0.639 | 0.759 | 0.810 | 0.828 | 0.830 | 0.780 | 0.798 |
| | W/ R | 0.468 | 0.374 | 0.607 | 0.717 | 0.274 | 0.1 | 0.383 | 0.411 | 0.418 | 0.576 | 0.779 | 0.809 | 0.820 | 0.710 | 0.819 | 0.814 |
| VE | | - | - | - | 0.629 | - | - | - | - | - | 0.602 | 0.598 | 0.610 | 0.696 | 0.732 | 0.704 | 0.736 |

Table 3: Results on Attitude Detection (AD) and Value Selection (VS). For deepseek-distill models, the models always output their thinking process (i.e. reasoning). Therefore, we only report reasoning results for those models "W/ R". For larger models like Deepseek-R1, o1, and o3, we report results on the same subset used for human evaluation, due to the high cost of running them on the full dataset. In the "Merged" setting, we merge similar options and in "Multiple" setting, story characters may hold different opinions. The "Merged" and the "Multiple" settings are both under reasoning settings.

- For value selection (VS), we employ a multi-step approach: (1) The model summarizes the topics mentioned in the story based on the provided options; (2) Selects values associated with the identified topics; (3) Reasons about which value best reflects the story; (4) Finally, outputs the selected value.

- For value extraction (VE), we ask the model to summarize the content for each topic and then predict relevant values based on the summarization.

### 4.3 CQ across Different LLMs

We first show the results of attitude detection and value selection in Table 3. Overall, larger models substantially outperform smaller models, and **summarize-then-analyse** prompting generally improves performance. Human participants achieve an average score of 0.689 and 0.765 on AD and VS respectively.

**The model struggles to detect nuanced attitudes beyond simple binary labels.** Although the AD task should be easier than value selection, its scores are even lower. One reason is that models struggle to distinguish neutral stances, such as *neither agree nor disagree*. While they can easily differentiate between *agree* and *disagree*, the presence of a neutral option can cause confusion. Even when a model correctly identifies *agree*, it may be distracted by the neutral choice and incorrectly select *neither agree nor disagree*. Additionally, models find it challenging to differentiate between varying levels of severity, such as *not often* and *not at all often*. Although it is also challenging for humans, humans can do better in identifying nuanced attitudes than models. We present results after merging options of varying levels of severity in Table 3.

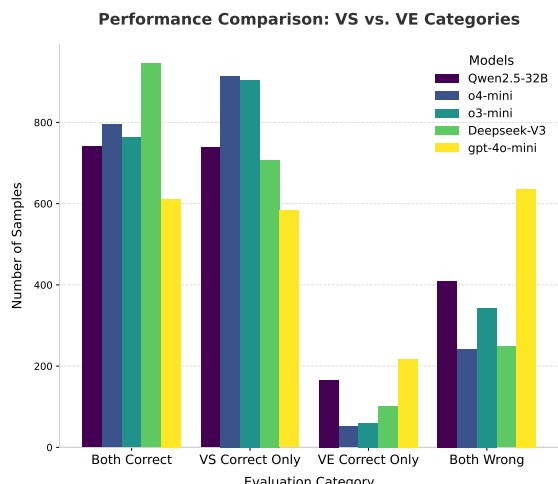

Figure 3: Comparison of VS vs. VE model performance on 5 different models.

**Smaller models fail in long CoT reasoning in cultural intelligence task.** Although CoT reasoning significantly enhances performance in large models, smaller models often struggle with long CoT reasoning. LLaMA models, in particular, perform poorly in adhering to CoT reasoning across both AD and VS tasks. While Qwen 7B follows CoT reasoning well in AD, its performance declines significantly in VS as the reasoning steps become longer. A manual inspection reveals that its final output often consists of random values or irrelevant phrases that fail to focus on the given options. The DS-distill models exhibit slightly better instruction-following capabilities, outperforming Qwen 7B and LLaMA 8B in VS. While they do not

Figure 4: Category specific results. Overall, models perform worst in the Religious setting, and category-specific datasets yield higher scores than randomly sampled ones.

strictly adhere to the prescribed format, their reasoning process generally aligns with the ideas provided in the prompt. However, their CQ reasoning ability remains weaker than their mathematical reasoning skills, resulting in final scores that are still lower than in the no-reasoning setting.

**Stronger models do not necessarily outperform weaker models in VE.** VE tasks require strong reasoning and summarization capabilities, and smaller models often struggle with this task, frequently producing nonsensical outputs. As a result, we focus our evaluation on five models: Qwen 2.5 32B, GPT-4o-mini, o3-mini, o4-mini, and DeepSeek-V3. We also evaluate a smaller subset of 25 stories using o1, o3 and DeepSeek-R1. Interestingly, unlike attitude detection and value selection—where larger models consistently outperform smaller ones—we observe that a weaker model (Qwen 2.5 32B) can outperform stronger ones like o3-mini and o1. One possible explanation is that current CoT reasoning methods are not well-suited for open-ended generation; models tend to perform better when given predefined options.

**Analysis of the Generative Gap: Discriminative vs. Generative Performance** We compare discriminative (VS) and generative (VE) tasks by classifying predictions into four groups: Both Correct, VS Correct Only, VE Correct Only, and Both Wrong. To provide a precise measure of reasoning capability, the counts in the figure reflect value counts rather than stories. Our analysis reveals a significant performance divergence between discriminative (VS) and generative (VE) tasks, despite identical input contexts. The data reveals a significant "Generative Gap": the "VS Correct Only" category remains substantial for most models, with counts reaching nearly 900 for o4-mini and Qwen2.5-32B. This indicates models frequently recognize values they fail to explicitly articulate. Conversely, the "VE Correct Only" category is consistently low, suggesting extraction is strictly harder than selection. After qualitative analysis, we identify two primary mechanisms driving this "Generative Gap":

- **The Abstractive Bottleneck:** Extraction tasks expose a critical dependency on semantic anchoring. Absent the specific options provided in multiple-choice settings (Selection), models revert to surface-level rephrasing of the explicit text. For instance, instead of identifying the normative ground truth *"On the whole, men make better business executives,"* the model extracted *"Gender imbalance leads to feelings of inadequacy."* While the latter accurately describes the character's emotional reaction within the specific story, it fails to capture the universal cultural axiom driving that friction. The models struggle to bridge the inference gap to latent, implicit cues without the scaffolding of a predefined answer space.

- **Safety-Alignment Distortion:** We detect a distinct interference from safety protocols. In instances involving biased or controversial character traits, models tend to "sanitize" the output—hallucinating fairness or reframing prejudice as traditionalism. For example, if the value is against gender equality, the extracted answer might instead express the opposite stance. This suggests the model's internal alignment constraints prioritize harmlessness over accurate cultural characterization during open-ended generation.

## 4.4 CQ Across Different Value Categories

We present results on category-specific datasets in Figure 4, showing the performance of six models in the no-reasoning setting. Complete results across all models and settings are provided in the Appendix B.1. Overall, the results show that models perform better in understanding political, social, and ethical values, achieving performance that is better than or comparable to the random setting. However, they perform worse in the religious values domain across both tasks. While models show stronger performance in attitude detection on the political dataset, they perform better in value selection on the ethical dataset. This suggests that it is easier for models to infer people's political stances but more challenging for them to identify the specific topics being discussed in political contexts. In contrast, in the ethics domain, models find it easier to identify

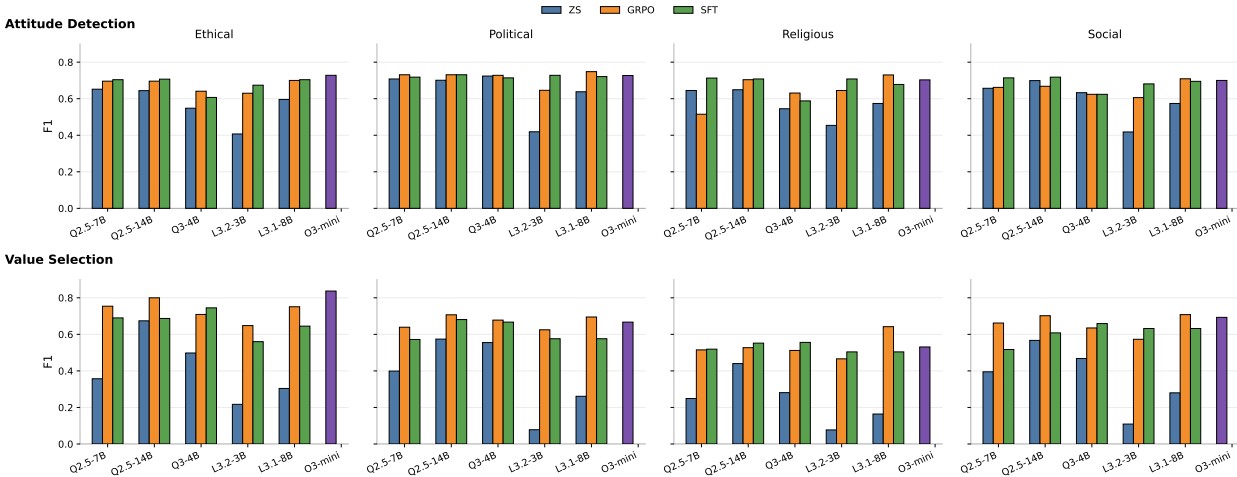

Figure 5: Category-specific results after fine-tuning smaller models with SFT and GRPO.

the main topics being discussed. For value extraction, overall, models perform better on category-specific results except for DeepSeek-V3, as they only need to cover a single topic.

# 5 Improving CQ in Small Models

In Section 4.3, we observe that smaller models struggle with long CoT reasoning, often failing to produce complete and coherent reasoning traces. We find that one-shot prompting does not notably improve performance (Appendix B.2). To strengthen smaller models' CQ reasoning ability, we explore two complementary fine-tuning approaches: **Supervised Fine-Tuning (SFT)** and **Reinforcement Fine-Tuning (RFT)**. Both aim to improve the model's ability to reason over cultural contexts but differ in learning strategy. SFT provides explicit reasoning supervision and encourages models to follow structured reasoning chains distilled from a larger model, while RFT promotes open-ended reasoning by allowing models to explore reasoning paths guided by reward signals.

## 5.1 Training Setup

For SFT, we distill reasoning traces from *o3-mini* using a random-setting dataset of 500 samples and fine-tune five smaller models (Qwen-2.5-7B, Qwen-2.5-14B, Qwen-3-4B, LLaMA 3.1-8B, and LLaMA 3.2-3B) with LoRA (Hu et al., 2021) for five epochs.

For RFT, we adopt the GRPO method Shao et al. (2024) to fine-tune smaller LLMs on both attitude detection and value selection tasks. For the attitude detection task, which is formulated as multiple-choice classification, the reward signal is binary—1 for correct predictions and 0 otherwise. In contrast, the value selection task is a multi-label setting, where predictions can be partially correct. To account for this, we define the reward as the ratio of true positives to the total number of predicted values. If the number of selected values differs from the ground truth, the reward is set to 0.

Since partial rewards tend to slow convergence compared to binary signals, we train models until performance stabilizes. Empirically, we find that five epochs are sufficient for the attitude detection task, while value selection typically requires around fifteen epochs to converge. We set the rollout number to 16 to ensure stable and fair gradient updates from group-based advantage estimation. We train the models on four A100 GPUs for 2–3 hours.

## 5.2 Results

Evaluation on category-specific datasets—which contain unseen values and thus approximate out-of-domain settings—reveals that SFT substantially improves reasoning quality and generalization across most domains. Notably, LLaMA 3.2-3B even surpasses *o3-mini* in political and religious AD tasks, while Qwen 14B exceeds it in the same domains for value selection (VS). GRPO achieves modest gains on the AD task and sometimes yields slightly lower results than SFT, likely due to its simpler multiple-choice nature. However, it generally yields larger improvements on the more challenging VS task, which requires complex reasoning, as shown in Figure 5. Overall, both SFT and RFT significantly outperform zero-shot prompting, particularly on tasks involving long CoT reasoning.

**Comparison between SFT models and zero-shot prompting**    To investigate the specific mechanisms by which Supervised Fine-Tuning (SFT) enhances reasoning quality, we conducted a manual review of 50 samples. Our analysis identifies three primary areas where SFT mitigates failures compared with zero-shot prompting:

- **Internal Inconsistency:** Discrepancies where the final answer ignores the preceding reasoning. For example, base models often discuss a specific topic (e.g., alcohol usage) throughout the reasoning but fail to include related values in the final output, or conversely, select values (e.g., regarding divorce) never mentioned in the logic chain.

- **Logical Fallacies:** Instances where the model links unrelated concepts. A notable example from a base model linked a preference for *"traditional family setups"* to stances on *"immigration priority for citizens."* SFT reduces these thematic "leaps" by enforcing stricter logical continuity.

- **Oversight of Granular Details:** Base models often rely on broad observations that miss crucial context. In one case, a base model generalized a discussion on animal ethics to *"hunting and the environment,"* whereas the SFT model correctly identified the specific nuance of *"using animals for entertainment"* (e.g., zoos/circuses), aligning accurately with the gold label.

The contrast between these reasoning patterns is summarized in Table 4. Our analysis shows that, by mimicking human-provided reasoning patterns, SFT exhibits emergent ability in overcoming certain reasoning failures. Although these improvements are not sufficient to fully resolve all issues, they demonstrate meaningful progress beyond simple pattern imitation.

| Category | Base Model Failure | SFT Improvement |
|---|---|---|
| **Consistency** | Logic discusses Topic A, but Answer selects Topic B. | Answer is strictly grounded in the preceding logic. |
| **Logic** | Creates false equivalence between unrelated social stances. | Maintains thematic and logical relevance. |
| **Granularity** | Generalizes specific topics (e.g., animal entertainment → hunting). | Captures precise details and context-specific keywords. |

Table 4: Comparison of reasoning patterns between Base and SFT models.

## 5.3 Qualitative Analysis: Divergent Cognitive Strategies in Cultural Reasoning

To understand the qualitative differences in how SFT and RFT models approach cultural reasoning, we analyzed the generated reasoning traces. The analysis reveals that while SFT models follow the reasoning path strictly and list detailed evidence, RFT attempts to *extract* the high-level cultural signal. Our qualitative analysis of reasoning traces reveals a fundamental divergence in cognitive strategies: while SFT models optimize for structural verifiability ("Glass Box"), RFT models optimize for abstractive efficiency ("Black Box"). The SFT model (distilled from `o3-mini`) rigorously tracks individual agents to build an evidence-based chain, often following a rigid template like *"First, Fatima states faith is private... Second, Jay's remark connects... Therefore, the value is Secularism."* This provides a clear audit trail but risks

"missing the forest for the trees" by over-analyzing local details. Conversely, RFT abandons agent tracking for a heuristic summarization, compressing the narrative into a thematic gist: *"The conversation discusses religious practice vs. public order... Conclusion: Secularism."* While this "direct extraction" is robust against conversational noise and effectively captures the high-level topic, its lack of specific citations renders the decision process opaque and difficult to debug, highlighting a clear trade-off between the interpretability of social simulation (SFT) and the robustness of semantic abstraction (RFT).

## 6 Related Work

**Culture-aware LLMs.** Culture-aware LLMs account for cross-cultural differences. Prior work has examined their cultural personas and consistency (Kharchenko et al., 2024; Rozen et al., 2024; Yao et al., 2024; Johnson et al., 2022; Saha et al., 2025b), showing that prompting language influences expressed values (Zhong et al., 2024). To improve cultural awareness, researchers have proposed both single- and multi-culture models through dataset augmentation and alignment (Nguyen et al., 2023; Lin & Chen, 2023; Abbasi et al., 2023; Li et al., 2024a; Wu et al., 2025). New benchmarks and datasets further support cultural knowledge acquisition (Myung et al., 2024; Shi et al., 2024), improving performance on tasks like hate speech detection (Li et al., 2024a).

**Value understanding.** Value understanding is key to effective human-LLM interaction. Prior work has focused on detecting general social norms in short texts, often using classification or entailment-based methods (Ren et al., 2024; Kiesel et al., 2023; Zhang et al., 2024; Li et al., 2023). Wu et al. (2025) introduced an English dataset with 3,060 human-crafted multi-turn scenarios across 60 countries and found that models struggle to interpret nuanced cultural contexts. Others have investigated datasets such as QA pairs, conversations, and story scenarios in Persian Monazzah et al. (2025), Japanese & Sundanese Pranida et al. (2025), and Korean Kim et al. (2024), reporting similar limitations in cultural and linguistic understanding. Fung et al. (2022) also introduced a method for extracting social norms from conversations, emphasizing norm mining rather than cultural value identification. In contrast, our work focuses on understanding cultural values within long, real-life conversations, contributing to the development of culture-aware LLMs in human-AI interaction.

## 7 Conclusion

We introduce **CQ-Bench**, a benchmark for evaluating whether LLMs can infer implicit cultural values from conversation. Unlike prior benchmarks that focus on short or explicit statements, **CQ-Bench** centers on long, multi-character conversations in which values are embedded indirectly in natural interaction. To support this goal, we construct a synthetic yet carefully validated dataset and evaluate models through three complementary tasks: attitude detection, value selection, and value extraction. Together, these tasks measure not only whether a model can recognize a value when given structure, but also whether it can recover values from open-ended conversational context.

Our results show that even strong frontier models continue to struggle with nuanced cultural understanding, especially when values are implicit, attitudes are subtle, or generation is required instead of selection. At the same time, the benchmark reveals meaningful variation across tasks and value categories, highlighting specific failure modes rather than a single aggregate weakness. We also show that fine-tuning on only 500 culturally rich examples can substantially improve smaller models, suggesting that cultural reasoning ability can be improved efficiently with targeted supervision.

Overall, **CQ-Bench** exposes current gaps in LLM cultural reasoning and provides a concrete foundation for future work on culturally intelligent AI. We hope the benchmark will support research not only on evaluation, but also on data construction, reasoning methods, and alignment strategies that enable LLMs to engage more responsibly and effectively with diverse human values and cultural contexts.

## Limitations

This study has several limitations. First, although we construct a multiple-attitude dataset, we evaluate only attitude detection on it; future work should study whether models can recover individual characters' values in conversation. Second, although we rewrite stories to remove explicit value expressions, rewrite quality remains uneven; detecting explicit speech is much easier than producing strong implicit alternatives. Third, models sometimes struggle with nuanced options during story generation, which can also make intended attitudes hard for humans to detect. Human CQ also varies widely, with some annotators reaching about 80% accuracy in attitude detection and others staying closer to 50%. Finally, GPT-4o was not reliable for automatically detecting reasoning flaws, so our qualitative error analysis relied on manual inspection of a small subset.

## Ethics Statements

All data used in this study were synthetically generated by large language models and do not contain any real user conversations or personal information. Cultural value statements were sourced from publicly available, anonymized survey instruments, including the World Values Survey and the GlobalOpinion dataset. Human participants were involved only in evaluating dataset quality and model outputs; they were not the subject of the study itself, which focuses on LLM behavior. We did not collect sensitive or personally identifying information from annotators.

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

# A  Dataset

## A.1  Value Set

We adopt seed values from the World Values Survey (WVS) and GlobalOpinion. The WVS is a global research project that explores people's values and beliefs and how they change over time. These values cover topics ranging from personal beliefs to political stances. The survey includes more than 200 values across 100 countries. Each value has a different set of candidate options, and we provide the possible option sets in Table 5.

| Example Value | Options |
|---|---|
| Do you think that your country's government should or should not have the right to do the following: Keep people under video surveillance in public areas? | Definitely should not have the right
Probably should not have the right
Probably should have the right
Definitely should have the right |
| In your view, how often do the following things occur in this country's elections: Journalists provide fair coverage of elections? | Very often
Not often
Not at all often |
| Work is a duty towards society. | Agree
Neither agree nor disagree
Disagree |
| Apart from weddings and funerals, about how often do you pray? | Frequently
Occasionally
Never |
| Having a strong leader who does not have to bother with parliament and elections. | Very good
Very bad |
| How important is it for people to help others? | Important
Not important |

Table 5: Example values and their options. "Definitely should not have the right" and "Probably should not have the right" are similar options with different levels of severity.

## A.2  Story Generation

For the random dataset, we start with 50 statements covering seven topics: social, migration, security, science and technology, religious, ethical, and political, following the categorization defined by Li et al. (2024a). To expand the dataset, we focus on four categories—social, religious, ethical, and political—because the WVS and GlobalOpinion datasets contain more values in these areas. To generate coherent stories, we require at least 20 statements per category. We manually select values from WVS and GlobalOpinion, excluding those that do not fit our setting (e.g., "How many times do you go to church every week—everyday"). As a result, we collect 27 seed statements for social values, 23 for religious values, 24 for political values, and 28 for ethical values.

For the multiple attitude dataset, we use the same 50 statements as in the random setting. Each story involves four characters, and we assign one value to each character. Compared to the random dataset, which contains 5 values per story, the multiple attitude dataset includes $5 \times 4$ values. Due to the increased value space, we only conduct attitude detection on the multiple attitude dataset, as value selection becomes challenging when the ground-truth set is already large.

For each story, we randomly predefine a scenario from the following locations: a company, a school, a neighborhood, a national park, a restaurant, an amusement park, and an airplane. We remove very short stories (less than 400 words). The length of stories ranges from 500 to 900 words.

| **Value:** One of my main goals in life is to make my parents proud. | |
|---|---|
| **Explicit Value Representation (Li et al. (2024b))** I firmly believe that one of my main goals in life has been to make my parents proud. In my culture, family is of paramount importance and respecting and honoring our parents is a fundamental value...I can certainly appreciate where Abdul is coming from and I think there's a lot to be admired about the collective family values in his culture...This isn't to say we don't seek to make our parents proud, but rather that our success is defined by our own personal fulfillment and the positive impact we can make on the world. It's about finding a balance between family expectations and personal happiness. | **Implicit Value Representation** That's always been a driving force for me. My parents made sacrifices for me, and I intend to honor that. My mom used to say that sticking by our community is like a badge of honor. I really want to honor her and show her that her values live on in me. When I see the smiles on my parents' faces after I accomplish something, it hits different. It's like all their hard work and sacrifices are acknowledged. |

Table 6: Value Representation Comparison

|  |  | Random | Political | Social | Religious | Ethical | Multiple |
|---|---|---|---|---|---|---|---|
| Word count | Original | 19.94 | 20.21 | 20.04 | 18.95 | 19.60 | 16.60 |
|  | Refined | 35.66 | 34.41 | 30.91 | 33.84 | 35.46 | 26.49 |
| Distinct-3 | Original | 0.877 | 0.878 | 0.876 | 0.860 | 0.875 | 0.850 |
|  | Refined | 0.941 | 0.939 | 0.939 | 0.937 | 0.940 | 0.919 |
| Distinct-4 | Original | 0.815 | 0.817 | 0.814 | 0.797 | 0.812 | 0.775 |
|  | Refined | 0.911 | 0.908 | 0.909 | 0.905 | 0.910 | 0.878 |
| Distinct-5 | Original | 0.755 | 0.755 | 0.752 | 0.742 | 0.750 | 0.702 |
|  | Refined | 0.882 | 0.878 | 0.878 | 0.874 | 0.880 | 0.837 |
| Similarity | Original | 0.510 | 0.488 | 0.574 | 0.742 | 0.508 | 0.491 |
|  | Refined | 0.433 | 0.408 | 0.473 | 0.355 | 0.415 | 0.425 |

Table 7: Statistics for explicit speech refinement show that the refined outputs exhibit greater linguistic diversity compared to the original speech.

### A.3 Story Validation

We conduct three validation stages: incorporation checking, consistency checking, and implicitness checking. We show the human annotation guidelines in Figure 6, 7, 8 and 9.

For the implicitness check, we compare the word count before and after rewriting to assess changes in length. We use Distinct-N (Li et al., 2016) to measure sentence diversity, which captures the number of distinct n-grams within a sentence. Finally, we compute semantic similarity using a Sentence Transformer (Thakur et al., 2021) between the value and both the original and rewritten speech. Ideally, the similarity score between the value and the rewritten speech should be lower, as the model is instructed not to mention the value explicitly. All statistics are reported in Table 7. The results show that refined speech is longer, more diverse, and semantically further from the value.

We generate 500 stories for the random setting and 100 stories for each of the remaining settings. Table 2 reports the number of attitude-detection datapoints and the total number of ground-truth values used for value selection.

We also conduct a human Turing-style evaluation to assess whether CQ-Bench utterances can be distinguished from naturally written human value expressions. We sample 100 annotation points and present annotators with one human-written utterance and one model-generated utterance in random order. For each pair, annotators choose among three options: *human*, *model*, or *tie*. A *tie* choice is assigned a score of 0, indicating that the annotator cannot distinguish between the two sources. For non-tie responses, the task is treated as binary classification, where the annotator must identify which utterance was written by

# CQ-Bench Annotation Guide (with Naturalness)

## 0) Goal (what you're judging)

We generate short, multi-speaker stories to reflect **five target cultural values**. Your job is to verify that the final story:

- reflects each value **implicitly** (not stated too bluntly),

- is **consistent** across speakers for each value,

- benefits from the model's **obviousness-reducing rewrites**, and

- reads like a **natural** human conversation.

    You'll see some **LLM "judge"** outputs (consistency/reflection validations). Treat them as **hints only**—your labels are the source of truth.

---

## 1) What each row contains (8 columns)

1. **Location** — where the story happens. *(You can ignore this.)*

2. **Values** — the five target values (V1–V5) the story aims to reflect.

3. **Consistent story** — the initial story drafted from the five values.

4. **Obvious check** — lines the LLM flagged as **too explicit** about a value.

5. **Obvious rewrite** — suggested rewrites to make flagged lines **more subtle**.

6. **Final story** — the story **after** applying the rewrites. Use this for your official judgments.

7. **Consistency validation (multiple)** — three LLM passes checking if **all characters** align on each value in the **Final story**.

Figure 6: Screenshot of human annotation guideline (1)

# CQ-Bench Annotation Guide (with Naturalness)

## 0) Goal (what you're judging)

We generate short, multi-speaker stories to reflect **five target cultural values**. Your job is to verify that the final story:

- reflects each value **implicitly** (not stated too bluntly),

- is **consistent** across speakers for each value,

- benefits from the model's **obviousness-reducing rewrites**, and

- reads like a **natural** human conversation.

  You'll see some **LLM "judge"** outputs (consistency/reflection validations). Treat them as **hints only**—your labels are the source of truth.

## 1) What each row contains (8 columns)

1. **Location** — where the story happens. *(You can ignore this.)*

2. **Values** — the five target values (V1–V5) the story aims to reflect.

3. **Consistent story** — the initial story drafted from the five values.

4. **Obvious check** — lines the LLM flagged as **too explicit** about a value.

5. **Obvious rewrite** — suggested rewrites to make flagged lines **more subtle**.

6. **Final story** — the story **after** applying the rewrites. Use this for your official judgments.

7. **Consistency validation (multiple)** — three LLM passes checking if **all characters** align on each value in the **Final story**.
8. **Reflection validation (multiple)** — three LLM passes checking if each value is **inferable** (implicitly present) in the **Final story**.

Figure 7: Screenshot of human annotation guideline (2)

# 2) Your tasks (A–D) and exactly what to submit

### A) Rewrite improvement (subtlety)

**What:** For every proposed rewrite in **Obvious rewrite**, decide if it improves subtle/indirect expression compared to the original line in **Consistent story**.

- If **not better**, label **0**.

- If **better**, rate **1–5** for *how much* better it is (1 = slightly; 5 = much better).

**Submit:** a comma-separated list in the **same order** as the rewrites appear.

- Example: 2, 0, 5, 3

- **If there are no rewrites for a story:** put NA.

  Judge only **implicitness/subtlety** (you may consider grammar if it affects subtlety). If a rewrite becomes *more explicit* or introduces contradiction, give **0** (or a low score).

---

### B) Consistency per value (V1–V5)

**Question:** In the **Final story**, do **all characters** align on each value?

- **1** = consistent (no character contradicts the value)

- **0** = inconsistent (at least one character contradicts/doesn't share it)

**Submit:** five comma-separated binaries in V1→V5 order, e.g., 1, 1, 1, 0, 1.

  If a character never mentions a value and nothing implies the opposite, treat as **consistent** unless the story suggests disagreement.

Figure 8: Screenshot of human annotation guideline (3)

## C) Reflection per value (V1–V5)

**Question:** In the **Final story**, is each value **implicitly reflected** (a careful reader could reasonably infer it without being spelled out)?

- **1** = reflected implicitly

- **0** = not reflected / not reasonably inferable

**Submit:** five comma-separated binaries in V1→V5 order, e.g., 1, 1, 1, 0, 1.

> "Implicit" means **no on-the-nose statements** like "I believe X because…". Look for actions, choices, tone, and dialogue cues.

---

## D) Conversational naturalness

**Question:** Does the **Final story** read like a natural human conversation?

**Consider:**

- **Turn-taking & flow:** turns respond to prior content; no abrupt topic jumps.

- **Persona consistency:** speakers' tone/stance fit their roles throughout.

- **Pragmatics:** hedges ("maybe," "I guess"), acknowledgments ("mm, makes sense"), concise lines rather than mini-lectures.

- **Non-redundancy:** avoids copy-paste feel or slogan repetition.

- **Readability:** minor typos are fine; penalize only if clarity or human-likeness suffers.

**Labeling:**

- **Score (1–5):** 1 = very unnatural, 2 = below average, 3 = acceptable, 4 = natural, 5 = very natural

- **Binary (0/1): 1** if natural (score ≥ 4), else **0**

Figure 9: Screenshot of human annotation guideline (4)

a human. This setup allows us to measure distinguishability while explicitly accounting for uncertainty in human judgments.

To collect naturalistic value-laden discourse from Reddit, we employ a two-phase pipeline. In Phase 1 (community discovery), we use the Reddit API via PRAW to search for relevant subreddits for each of the seven WVQ topic categories, such as social values, migration, and security. For each topic, we issue multiple short keyword queries (e.g., gender equality, immigration, and public safety) to Reddit's subreddit search endpoint, filter the results to non-NSFW communities with at least 10,000 subscribers, and retain the top 20 communities per topic ranked by subscriber count. The discovered communities are then manually reviewed before proceeding to the next stage. In Phase 2 (speech extraction), for each WVQ value question, we construct topic-specific search queries and search the discovered subreddits for matching posts and top-level comments. We apply a tiered quality-filtering scheme—strict, relaxed, and fallback thresholds—based on post or comment score, minimum text length, and keyword-match count to retain substantive and opinion-bearing content. Extracted utterances are capped at 40 per value question and 2,000 in total, deduplicated, and stored in JSONL format together with metadata such as subreddit, score, matched keywords, and the associated WVQ question ID. We then use an LLM as a judge to verify whether the extracted speech matches the target value, after which we manually remove low-quality or mismatched examples. To reduce privacy risks, we replace explicit named entities such as politician names or company names with generic descriptions (e.g., replacing Bloomberg with media company). The Reddit corpus is used only for the human Turing test, is not included in the benchmark release, and will not be publicly distributed.

| Value | Original Speech | Rewritten Speech |
|---|---|---|
| In your view, how often do the following things occur in this country's elections: Voters are bribed?–Very often | Speech: Voter bribing happens so often , it feels like a norm at this point. Explanation: The speech basically rephrases the value without providing supporting evidence | But honestly, I wasn't surprised. You hear about the rumors and tales around voting—sometimes it seems like it's just part of the landscape now. |

Table 8: An example of implicitness check and rewriting.

| | | Qwen | | Deepseek-distill | | | GPT | Deepseek |
|---|---|---|---|---|---|---|---|---|
| | | 14B | 32B | Q 1.5B | Q 7B | L 8B | 4o-mini | V3 |
| AD | Political | 0.734 | 0.741 | 0.455 | 0.558 | 0.631 | 0.754 | 0.781 |
| | Social | 0.699 | 0.699 | 0.277 | 0.239 | 0.456 | 0.718 | 0.709 |
| | Ethical | 0.719 | 0.719 | 0.437 | 0.474 | 0.637 | 0.711 | 0.533 |
| | Religious | 0.649 | 0.631 | 0.431 | 0.506 | 0.637 | 0.592 | 0.574 |
| VS | Political | 0.574 | 0.587 | 0.372 | 0.206 | 0.402 | 0.56 | 0.694 |
| | Social | 0.567 | 0.618 | 0.305 | 0.314 | 0.323 | 0.568 | 0.715 |
| | Ethical | 0.674 | 0.741 | 0.366 | 0.341 | 0.389 | 0.64 | 0.815 |
| | Religious | 0.390 | 0.466 | 0.362 | 0.355 | 0.376 | 0.398 | 0.516 |

Table 9: Results on category-specific dataset under zero-shot reasoning setting.

# B Results

## B.1 Category-Specific Results

The no-reasoning results are shown in Figure 4. Additional category-specific results are summarized in Table 9. The social-category dataset uses a single set of options: agree, disagree, and neither agree nor disagree. To study how well models understand the middle stance, we first remove questions for which the ground truth is neither agree nor disagree, which eliminates about one-third of the data. We then remove the neither

agree nor disagree option from the remaining examples, resulting in a fully binary dataset. On this binary version, o3-mini achieves an accuracy of 0.911. When we reintroduce the neither agree nor disagree option, performance drops to 0.811. However, when evaluated on the full dataset—including questions with neither as the correct answer—o3-mini achieves only 0.700 accuracy. This suggests that including a middle stance significantly challenges the model's judgment.

## B.2 Distillation on Smaller Models

We use reasoning traces generated by o3-mini to fine-tune smaller models. We use the LLaMA-Factory framework Zheng et al. (2024) and LoRA to accelerate fine-tuning Hu et al. (2021). We train for five epochs on a single A6000 GPU for 2–3 hours. The LoRA rank is 8, and the learning rate is 0.0001. We evaluate multiple hyperparameter combinations before selecting the final setting. All experiments are single runs.

We find that reasoning without fine-tuning has several recurring issues. The first is inconsistency: for example, the reasoning may mention alcohol use multiple times, but the final answer fails to include values related to alcohol. In other cases, the reasoning does not mention divorce at all, but the final answer chooses values related to divorce. The second issue is logical error, for example: *"Oliver's preference for a traditional family setup for raising kids matches When jobs are scarce, employers should give priority to people of this country over immigrants.–Neither agree nor disagree."* There is no direct logic between two statements. The third is overlooking details. For example, the one-shot reasoning is *"The discussion also involves ethical considerations regarding the use of animals and the environment, which is reflected in the option: Hunting animals as a sport should be banned. –Disagree."* The reasoning itself sounds reasonable, however, the gold label is *"Using animals for entertainment like in zoos/circuses is ethical.–Disagree."* The SFT reasoning is *"Finally, Harold's statement on **using animals for entertainment**, which he finds shocking, aligns with the value that such practices are unethical.* The difference between the two reasoning traces is that the latter identifies the specific detail of **using animals for entertainment**, while the former mentions only the broader theme of animals and the environment.

## B.3 Ablation Study

We conduct ablation studies using the **Llama 3.2 3B** model on both the *Attitude Detection (AD)* and *Value Selection (VS)* tasks. We fine-tune the model with {50, 100, 250, 500} stories, both with and without chain-of-thought (CoT) reasoning. Each configuration is trained with three random seeds, and we report the mean and standard deviation. To examine model size effects, we also evaluate the **Qwen 2.5 14B** model on the VS task.

| Stories | Setting | Religious | Social | Ethical | Political | Multiple |
|---|---|---|---|---|---|---|
| 50 | W/O CoT | 0.5474 (0.0061) | 0.6369 (0.0058) | 0.5025 (0.0126) | 0.6965 (0.0095) | 0.5835 (0.0043) |
| | CoT | 0.5537 (0.0156) | 0.6761 (0.0101) | 0.5074 (0.0212) | 0.6910 (0.0141) | 0.5729 (0.0072) |
| 100 | W/O CoT | 0.6737 (0.0062) | 0.6651 (0.0089) | 0.6655 (0.0046) | 0.6921 (0.0063) | 0.5903 (0.0060) |
| | CoT | 0.6510 (0.0125) | 0.6636 (0.0059) | 0.6308 (0.0122) | 0.6855 (0.0063) | 0.5714 (0.0105) |
| 250 | W/O CoT | 0.6792 (0.0099) | 0.6917 (0.0059) | 0.6951 (0.0182) | 0.6955 (0.0042) | 0.5617 (0.0117) |
| | CoT | 0.6463 (0.0111) | 0.6479 (0.0102) | 0.6383 (0.0257) | 0.6689 (0.0110) | 0.5615 (0.0052) |
| 500 | W/O CoT | 0.6863 (0.0073) | 0.7232 (0.0042) | 0.6716 (0.0087) | 0.6885 (0.0022) | 0.5632 (0.0035) |
| | CoT | 0.6820 (0.0190) | 0.6918 (0.0079) | 0.6728 (0.0076) | 0.7178 (0.0142) | 0.5623 (0.0065) |

Table 10: Ablation study on Attitude Detection using Llama 3.2 3B. We report average accuracy and standard deviation across three seeds.

We summarize key findings as follows:

- **Impact of dataset size.** In attitude detection, increasing dataset size does not strongly correlate with performance, likely because the task involves only two to three attitude options and the pat-

| Stories | Setting | Religious | Social | Ethical | Political |
|---------|---------|-----------|--------|---------|-----------|
| 50 | W/O CoT | 0.3204 (0.0070) | 0.3673 (0.0177) | 0.4355 (0.0132) | 0.3571 (0.0339) |
|    | CoT | 0.2849 (0.0035) | 0.2921 (0.0145) | 0.3326 (0.0074) | 0.3116 (0.0116) |
| 100 | W/O CoT | 0.3312 (0.0082) | 0.4515 (0.0065) | 0.5080 (0.0666) | 0.4635 (0.0147) |
|    | CoT | 0.3606 (0.0359) | 0.4249 (0.0191) | 0.5003 (0.0264) | 0.3837 (0.0016) |
| 250 | W/O CoT | 0.4668 (0.0263) | 0.5429 (0.0119) | 0.6525 (0.0100) | 0.5666 (0.0057) |
|    | CoT | 0.4172 (0.0128) | 0.5300 (0.0051) | 0.6304 (0.0132) | 0.5263 (0.0271) |
| 500 | W/O CoT | 0.4881 (0.0113) | 0.5859 (0.0134) | 0.7076 (0.0147) | 0.5957 (0.0110) |
|    | CoT | 0.4036 (0.0059) | 0.5672 (0.0170) | 0.6089 (0.0100) | 0.5181 (0.0029) |

Table 11: Ablation study on Value Selection using Llama 3.2 3B. We report average accuracy and standard deviation across three seeds.

| Stories | Setting | Religious | Social | Ethical | Political |
|---------|---------|-----------|--------|---------|-----------|
| 50 | W/O CoT | 0.4272 (0.0156) | 0.6093 (0.0087) | 0.6916 (0.0079) | 0.5625 (0.0154) |
|    | CoT | 0.4642 (0.0091) | 0.5971 (0.0179) | 0.6758 (0.0093) | 0.5996 (0.0021) |
| 100 | W/O CoT | 0.4210 (0.0167) | 0.5909 (0.0067) | 0.6902 (0.0179) | 0.5708 (0.0062) |
|    | CoT | 0.4951 (0.0109) | 0.5876 (0.0101) | 0.6720 (0.0137) | 0.5634 (0.0251) |
| 250 | W/O CoT | 0.4099 (0.0157) | 0.6002 (0.0060) | 0.6794 (0.0117) | 0.5821 (0.0149) |
|    | CoT | 0.5461 (0.0076) | 0.5868 (0.0148) | 0.6697 (0.0148) | 0.5962 (0.0082) |
| 500 | W/O CoT | 0.4817 (0.0069) | 0.6129 (0.0067) | 0.7672 (0.0043) | 0.6663 (0.0032) |
|    | CoT | 0.5465 (0.0122) | 0.6991 (0.0148) | 0.7524 (0.0200) | 0.6935 (0.0039) |

Table 12: Value Selection results on Qwen 2.5 14B.

tern is quickly learnable. In contrast, value selection shows consistent improvement as dataset size increases, especially from 50 to 250 samples.

- **Effect of CoT reasoning.** Surprisingly, non-reasoning (W/O CoT) fine-tuning often outperforms CoT in the Llama 3.2 3B model, particularly in the VS task. Manual inspection reveals that smaller models struggle to maintain coherent reasoning chains, often producing disorganized outputs.

- **Effect of model size.** For the Qwen 14B model, CoT begins to show advantage as dataset size increases, indicating that larger models can better leverage reasoning supervision. While the improvements are not universally significant, results suggest that CoT becomes beneficial only beyond a certain data scale and model capacity.

- **Alignment with prior findings.** Our observations are consistent with prior work, which shows that CoT mainly benefits symbolic or mathematical reasoning tasks, and may not generalize as effectively to social or value-based reasoning.

### B.4 Case study on value expression

To better assess the model's ability to express values implicitly, we conduct a small case study using Qwen 2.5 7B. The model is prompted to respond to an ongoing conversation given a target value, with the goal of reflecting the value implicitly while maintaining natural conversational flow. We compare the vanilla Qwen 2.5 7B model with its GRPO-tuned counterpart.

To evaluate the quality of generated speech, we employ an LLM-as-a-judge framework that assesses three dimensions: value reflection, speech naturalness, and value implicitness. We experiment with two judging strategies: (1) pairwise comparison, where the LLM directly compares two speeches, and (2) independent rating, where it scores each speech separately before comparison. Across both evaluation methods, the GRPO-tuned model performs slightly better, with approximately 5% of speeches showing improvement,

though the gain is not statistically significant. This case study suggests that while GRPO tuning enhances implicit value expression modestly, further improvements could be achieved by jointly strengthening detection and expression capabilities.

## C Human Annotations

To compare human performance with LLMs, we recruit several participants from computer science undergraduate and master's programs as annotators. The participants come from different backgrounds, including two Chinese students and two Indian students. We provide them with the same prompts used for the LLMs.

# D  Prompts

In this section, we present all prompts used for story generation, experiments, and validation.

---

**Story generation prompt**

##Identity
You are an expert in culture understanding.
## Task
You will be provided with 5 cultural values and a location where the conversation happens. Each of them follows the format [culture]–[value]. The first [culture] describes a statement or a situation, and [value] is how you agree with the culture or is the culture common or not.
Your task is to generate a scene including conversations and actions among multiple people and the scene needs to reflect the culture values provided.
Here are some requirements of the scene:
1. It cannot be too short. It should have multiple rounds of interaction among people.
2. It should not be too obvious. It cannot directly spit out or rephrase the values.
3. It cannot be easy to human to understand the culture values behind.
4. You do not need to follow the order of the values. You could mention the values multiple times through the conversation. Make sure the conversation flows well.
5. All the characters should follow the given values. There should not be contradictions between the character's value and the given value.
##Input
Here are the cultural values you should follow when generating: values
Here is the pre-defined location of the scene: location
##Output Now using the cultural values to generate a story.

---

**Incorporation check prompt**

##Identity
You are an expert in culture understanding.
##Task
You will be provided a story and values reflected in the story. Your task is to check if the story reflects values? The story does not mention the values directly. You will need some reasoning to analyze the story.
##Input
Here is the original story: story
Here are the values to reflect in the story: values
##Output
For each value, output if the value is reflected and provide reasoning. In the end, output the values not reflected without the reasoning. Only output the exact and comprehensive value including "–" within it, do not rephrase! If all the values are reflected, just leave it blank. Follow the format: [Value]:[Reasoning, Yes/No] ..... Values not reflected: [Value]

**Missing values incorporation prompt**

##Identity
You are an expert in culture understanding.
##Task
You will be provided a story and values which need to be reflected in the story.
Your task is to refine the story to reflect the value provided. You cannot remove anything or replace existing speeches from the story, you can only add conversations to reflect the value.
The refinement should flow with original story well. You cannot add new conversation randomly.
##Input Here is the original story: story
Here are the values to reflect in the story: values
##Output Now refine the story.

**Consistency check prompt**

##Identity
You are an expert in culture understanding.
##Task You will be provided a story and values reflected in the story. Your task: for each value, check if all the characters agree with the value. If there is characters who does not agree with the value, you should output the character's name and his speech, and why the speech does not align with the value.
##Input Here is the original story: story
Here are the values to reflect in the story: values
Now check the story and output if there is any contradiction. You can output reasoning to help you analyze. However, in the end, only output where is the contradiction one by one. If there isn't a contradiction, just reply NO. Otherwise, reply where is the contradiction.
##Output Follow the format strictly, do not change the format, output exact values from the values provided, and do not rephrase:
[Reasoning]:
[Value–attitude]:[If all the speeches are aligned with the value]
[Contradictions]:
[Value–attitude]: [*character name*:speech] ......

**Consistency resolve prompt**

##Identity
You are an expert in culture understanding.
##Task
You will be provided a story and values which need to be reflected in the story. However, the story includes some contradiction where characters do not agree on certain values. You will be provided where is the contradiction.
The contradiction includes 3 parts:
1. Correct value to follow
2. Character name
3. Character speech
Your task is to replace the speech mentioned in the contradictions with a new speech to make sure the speech is aligned with the values The refinement should flow with the original story well. You cannot add new conversations randomly.
##Input Here is the original story: story
Here are the contradictions: contradiction
Ignore the original character's speech. Directly write a new speech that reflects the value.
##Output Here is the rewritten speech:

**Implicitness check prompt**

##Identity
You are an expert in culture understanding.
##Task You will be provided a story and values reflected in the story. Your task is to check if there is obvious speech that directly mentions or rephrases the values. If the story mentions phrases or sentences from values, that would be also counted as directly mentioned If it just reflects the value but does not rephrase the values, it is not considered as obvious speech. Here are some examples of obvious speech:
[Example 1]
Value: In your view, how often do the following things occur in this country's elections: Journalists provide fair coverage of elections—Very often Speech: "I think journalists do their best to cover things fairly" Explanation: The speech basically rephrases the value without providing supporting evidence
[Example 2]: .......
You should output places where the values are directly mentioned. If the values are not directly spit out, do not output them!!!
##Input Here is the original story: story
Here are the values to reflect in the story: values
##Output Now check if any parts of the stories directly mentioned the values. You can output reasoning or evidence and help you analyze. However, in the final answer, output speeches where the values are directly mentioned and the corresponding values. If there are no directly mentioned values, just output NO in the final answer.
Follow the format:
[Reasoning]: reasoning
[Final answer]: [value]:[speech]
[value]:[speech] .....

**Implicitness rewrite prompt**

##Identity
You are an expert in culture understanding.
##Task
You will be provided a story and values reflected in the story. You will be also provided with some comments where the values are directly mentioned in the story. Each comment includes a value and a speech where the speech directly reflects the value. Your task is to refine the speech by rewriting the places where they directly mention the values. Instead of speaking out values directly, characters should reflect values implicitly by using anecdotes. Imagine the place is not a safe place to speak out, characters need to respect each other and not offend others. You need to make sure after rewriting, people are able to detect values by the speech. It cannot be too abstract.
Here are some examples:
[Example 1]: Value: How frequently do the following things occur in your neighborhood: Robberies?– Not frequently Original speech: "Well, at least we're not dealing with robberies or anything in our area." Refined speech: "You know in other counties, cars got breaking in is common. At least we don't have to worry about that."
[Example 2]: .....
##Input Here is the original story: story
Here are the values to reflect in the story: values
Here are the comments that you need to refer to when refining the story: comments
##Output
Please rewrite the speech one by one in the story. Do not miss any comments.
For each rewritten speech, you need to follow this format: Value:[value] Original speech: [Speech in the comment] Refined speech: [You rewritten speech]
Here is your output:

**Attitude detection prompt**

##Identity
You are an expert in culture understanding.
##Task
You will be given a scenario including background, conversation and actions in a location. You task is to detect the culture values reflected from the conversation. You will be provided a statement and you need to detect what attitude they hold to the value. For example, the statement is "The people who don't work turn lazy" and the options are "disagree, agree, neither disagree nor agree". And you need to read the story and choose one attitude from the options. The options could also describe something happening around them. For example, if the statement is "How often does this happen in your neighborhood:Racist behaviors", the options are "Very often, not often, not at all often", you need to choose one based on the story.
##Input Here is the scenario: story
Here is the statement: statement
Here are the options: options
##Output Now choose one option based on the story. Let's think step by step! First, extract the speech that is related to the statement.Then analyse the attitude of those people from the speech. Finally, output the answer.
Follow the format:
[Related speech]: speeches
[Analysis]: analysis of attitude
[Answer]:answer

**Value selection prompt**

##Identity
You are an expert in culture understanding.
##Task You will be given a scenario including background, conversation and actions in a location. You task is to detect the culture values reflected from the conversation.
You will be provided 15 options and you need to select number correct answers from the options. (Only number options are correct).
You need to choose the whole option, for example:
Being a housewife is just as fulfilling as working for pay–agree
Make sure you include '–' in the answer.
##Input
Here is the scenario: story
Here are the options: options
## Instruction To think step by step: 1. You need to first detect what topics the story mentioned according to the options. You need to list the speech related to the topic. 2. Then for each topic, detect what values might be related to the topic. Sometimes, you might find multiple options that might be correct, for uncertain options, you should compare them. 3. Based on the previous detection, analyze what values are aligned with the story, you should always prioritize those values that are strongly related. 4. In the end, output the final answers only. Only choose the required number of values.
##Output Follow the format: [Topic]: topic1: speeches related to the topic .....
[Value detection]: topic1: what values are related to the topic
[Reasoning]: Reasoning
[Final answer]:
[text] – [text]
.....
"""

**Value extraction prompt**

##Identity
You are an expert in culture understanding.
##Task
You will be provided with a story. Your goal is to identify the most prevalent topic values in the story. For example, one religious value could be: I strongly believe in God and the afterlife. The value should be a complete sentence, it should not be a phrase like work-life balance. The value should hold a attitude or it reflects social phenomena, it should not be a overview of topic. For example, one social value could be: I think work is a duty towards the society. And one political value could be: I think voters are bribed in our election system. Please identify the values in this story: story by paying attention to how the characters in the story discuss topic. You need to generate 10 values in total, make sure values are specific and detailed. Only focus on one topic in one value. Do not mention several themes in one value.
##Input Here is the story: story
Here is a story summary with some key ideas: summary.
##Output
You could output reasoning before you output final answer. But in the end, your output should follow the format: [Final answer]: value1 value2 ....

**LLM evaluation prompt**

##Identity
You are an expert in culture understanding.
##Task
You will be given two sets of texts: a set of predicted values and a ground truth set of values. Your task is to determine how many of the ground truth values are fully represented in the predicted values. A ground truth value is considered correct if all of its components are meaningfully discussed in the predicted value, even if there is no exact 1-to-1 match. It could be many-to-1 match i.e. many values to 1 ground truth value. If the ground truth value is fully presented, score 1, if it is partially presented, score 0.5, if it is not mentioned at all, score 0. Additionally, provide a brief justification for your score, explaining which values were correctly or incorrectly represented, in the justification, you should explicitly mention which predicted values are related to the ground truth value.
##Input
Here are the predictions: pred and ground truth: gt In the reasoning, if the ground truth value is fully represented, you need to point out which predicted value is related to it.
##Output
Return the results in this format: [Reasoning]:
[Ground truth 1]: [reasoning]
.....
[Final answer]:
[Ground truth 1]:1
[Ground truth 2]:0.5
......

