# OpenReview forum: "Can LLMs Grasp Implicit Cultural Values? Benchmarking LLMs' Cultural Intelligence with CQBench"
_TMLR — Under review for TMLR_

### Review · Reviewer_iLru · 2026-05-20

**Summary Of Contributions:**

The paper introduces CQ-Bench, a benchmark for testing whether LLMs can infer implicit cultural values from multi-character conversations. It uses value statements from World Values Survey (WVS) and GlobalOpinion and defines three tasks: attitude detection, value selection, and value extraction.

**Audience:**

Yes

**Audience Explanation:**

The paper studies an important and timely problem: whether LLMs can understand implicit cultural values in realistic conversations. This is relevant to researchers working on LLM evaluation, cultural alignment, value understanding, responsible AI, and synthetic benchmark construction. The three-task setup also makes the benchmark useful for analyzing different levels of cultural reasoning.

**Broader Impact Concerns:**

The paper addresses an important responsible-AI problem, but the broader impact discussion should be expanded. A benchmark for inferring cultural values from conversation can be useful for improving culturally sensitive systems, but it also raises risks. First, models trained or evaluated on this task may over-infer sensitive beliefs from limited conversational evidence. In real deployments, this could lead to stereotyping, profiling, or inappropriate personalization.

**Claims And Evidence:**

Yes

**Claims Explanation:**

The main claims are generally supported. The paper provides a clear benchmark design, describes the dataset construction pipeline, includes human validation, and evaluates many open-source and closed-source LLMs. The results support the claim that LLMs struggle more with nuanced attitude detection and open-ended value extraction than with value selection.

**Requested Changes:**

1. Add confidence intervals or significance tests for the main results. Many claims compare models with differences of only a few F1 points. The paper should report bootstrap confidence intervals over stories or values, especially for claims such as smaller fine-tuned models outperforming stronger models.

2. Reduce dependence on GPT-4o as both validator and judge. GPT-4o is used in several places: validation, consistency checking, and VE judging. The authors should add at least one independent judge model, more human evaluation for VE, or a disagreement analysis between judges. Otherwise, the benchmark risks measuring alignment with GPT-4o's cultural interpretation rather than model-independent cultural reasoning.

3. Clarify train/test separation for fine-tuning. The paper should explicitly state whether training and evaluation differ in values, scenarios, characters, templates, and generator prompts. If only values are unseen but scenarios or stylistic templates are shared, the reported generalization may be overestimated.

4. Reconsider the claim that CQ-Bench captures realistic human value expression based on the Reddit Turing-style test. The reported 44.8% accuracy from only three annotators is not enough to conclude strong realism. Below-random accuracy may indicate annotator confusion, imbalance, unclear instructions, or artifacts in the task design. The authors should provide sample size, confidence intervals, balancing details, and examples.

5. Add qualitative examples of failure cases for each task. The paper would benefit from concrete examples where models confuse neutral vs. agree, severity levels such as "not often" vs. "not at all often", or extract a sanitized value instead of the actual biased value.

6. Discuss cultural and linguistic coverage more carefully. The benchmark is based on WVS and GlobalOpinion values, but the conversations appear to be English synthetic dialogues. The paper should avoid implying broad global cultural coverage unless multilingual and region-specific settings are evaluated.

7. Improve writing precision. Some claims are stronger than the evidence supports, especially around dataset realism, human-level performance, and model grasp of implicitness. These should be softened or backed with additional experiments.

---

### Review · Reviewer_5mqS · 2026-06-12

**Summary Of Contributions:**

The paper introduces CQ-Bench, a benchmark for evaluating whether LLMs can infer implicitly expressed cultural values from multi-character conversations. Value statements are drawn from the World Values Survey and the GlobalOpinion dataset and embedded into synthetic dialogues via an LLM-based generation-and-validation pipeline with human review. The benchmark defines three tasks (attitude detection, value selection, and value extraction) and the authors evaluate a range of open- and closed-source models, plus SFT and GRPO fine-tuning of smaller models.

**Audience:**

Yes

**Audience Explanation:**

The benchmark and dataset are a useful contribution that researchers working on cultural alignment, value evaluation, and LLM benchmarking will likely find worth knowing about. A few findings also stand out, such as the gap between value selection and value extraction and the difficulty of the religious category.

**Broader Impact Concerns:**

The paper includes an Ethics Statement, which adequately covers data and participant concerns. However, it does not address the dual-use dimension of the work itself. The capability the paper studies and improves — generating and recognizing cultural values expressed implicitly and in a natural, human-like way — is most directly applicable to models impersonating humans, with the associated risks of fraud and deceptive use. I would ask the authors to extend the statement to address the intended use of this work and how this misuse risk should be understood.

**Claims And Evidence:**

No

**Claims Explanation:**

The broad empirical evaluation across model families and the ablation studies are credible, and they support the main descriptive findings. My concern is that the paper's presentation is confusing and the headline numbers tend to oversell what the actual measurements show.

The abstract and text lean on impressive-sounding figures like 94.5% agreement and describe the model as having a strong grasp of implicitness, but the underlying numbers are more modest: the Cohen's κ values are 0.546 for value reflection and 0.515 for consistency, both indicating moderate agreement; the naturalness average (3.63) sits between "acceptable" and "natural" on the authors' own scale; and the implicitness-improvement average (3.28) indicates only moderate improvement. These are framed more strongly than they warrant.

Some claims also can't be checked as presented: the Section 5.2 comparisons (e.g., small fine-tuned models surpassing o3-mini in certain domains) rest only on Figure 5, where the categories look nearly identical and no companion table is given; Figure 4 appears not to have rendered in the PDF I reviewed (please verify); and the dataset counts are reported confusingly, with 150 human-validated stories never clearly reconciled against the 1,000 generated.

**Requested Changes:**

Major:
- Consolidate the dataset numbers in one place, using tables and percentages where helpful. In particular, reconcile the 150 human-validated stories against the 1,000 generated, and state how those 150 were selected from the full set.
- Report the validation and quality numbers in a way that matches what they show. The 94.5% agreement is presented as strong, but the Cohen's κ values (0.546 for value reflection, 0.515 for consistency) are moderate, and the 1–5 naturalness (3.63) and implicitness-improvement (3.28) averages are middling. These should be framed accordingly rather than as strong evidence.
- Fix Figure 4, which appears not to have rendered.
- Make the Section 5.2 results verifiable: Figure 5 is hard to read (the categories look nearly identical), and the comparisons claimed in the text (e.g., small fine-tuned models surpassing o3-mini in certain domains) need a companion numeric table.

Minor / easy fixes:

- The claim that CQ is "crucial" for LLMs has no justification; support or soften it.
- The Section 5 characterization of how SFT and RFT differ is stated without evidence or citation and should be omitted, since the Glass Box / Black Box discussion in Section 5.3 already covers this.
- Add a Broader Impact discussion addressing the dual-use of this work toward impersonation or fraud (See below).

---

> ### Author Response · Authors · 2026-07-21
>
> **Response to Reviewer**
>
> Thank you for your constructive feedback. Please find our responses to your concerns below.
>
> **1. Dataset presentation and validation coverage.**
> We will consolidate all datasets into a single table in the revision. Regarding the human validation set: the validated samples were randomly drawn from the 500 generated conversations, and we ensured that they cover all topics present in the full dataset.
>
> **2. Human annotation agreement.**
> We appreciate your concern about the agreement scores. A key reason the original scores appear low is the highly imbalanced label distribution: the vast majority of labels are 1 and very few are 0. Under such imbalance, chance-corrected metrics like Cohen's kappa are known to penalize even small amounts of disagreement severely (the "kappa paradox"). We therefore recomputed agreement using Gwet's AC1, which is robust to label imbalance. The resulting scores are **0.871** for reflection and **0.817** for consistency, indicating strong agreement.
>
> On naturalness (3.63), we note that in our rubric, 3 = *acceptable*, 4 = *natural*, and 5 = *very natural*, so this score indicates that annotators on average judged the conversations as falling between acceptable and natural. Achieving fully human-indistinguishable dialogue is inherently difficult for LLM-generated conversations (e.g., they rarely contain the disfluencies and grammatical errors typical of human chat), but most annotators rated our conversations above the acceptability threshold.
>
> Similarly, for implicitness improvement (3.28), the rubric assigns 0 = *no improvement* and 1–5 = *increasing degrees of improvement*. A mean of 3.28 therefore reflects substantial improvement rather than a middling result.
>
> **3. Claim about CQ.**
> We will add supporting references for this claim in the introduction and make the justification explicit in the revision.
>
> We will address all remaining points in the revision as well. Thank you again for your thoughtful review!

---

### Review · Reviewer_gusa · 2026-07-03

**Summary Of Contributions:**

This paper introduces CQ-Bench, a synthetic benchmark for evaluating whether LLMs can infer implicit cultural values from multi-character conversations. The benchmark is built from WVS and GlobalOpinion values and includes three tasks: attitude detection, value selection, and value extraction. The authors propose an LLM-based data generation and validation pipeline with incorporation, consistency, and implicitness checks, followed by human validation. They evaluate multiple open- and closed-source LLMs and further study whether SFT/GRPO can improve smaller models on the benchmark.

Strengths:

1. The paper studies an important and underexplored problem: implicit cultural value understanding in conversational contexts.
2. The three-task design is reasonable and useful, moving from attitude classification to candidate value selection and open-ended value extraction.
3. The benchmark construction pipeline includes multiple validation steps, and it is good to see some human validation rather than relying entirely on automatic checks.
4. The empirical analysis reveals useful failure modes, such as difficulty with nuanced attitudes, religious values, and open-ended extraction.

Weaknesses:

1. The benchmark is almost entirely LLM-generated, LLM-revised, and LLM-validated, which raises concerns about synthetic artifacts and GPT-specific bias.
2. The human validation effort is helpful but limited, with only moderate agreement in some dimensions.
3. It is unclear whether performance on CQ-Bench transfers to real human conversations, since Reddit data are used only for distinguishability testing rather than as a labeled benchmark subset.
4. The results are not tested for sensitivity to the prompt and evaluation harness and more model family.
5. The SFT improvements may reflect learning benchmark-specific style and label patterns rather than general implicit cultural reasoning ability.

**Audience:**

Yes

**Audience Explanation:**

The paper addresses an important topic for LLM evaluation, cultural alignment, and social reasoning. The three-task setup is useful, and the findings on nuanced attitudes, religious values, open-ended extraction, prompt sensitivity, and small-model fine-tuning would interest researchers working on benchmark construction and culturally aware LLMs.

**Broader Impact Concerns:**

The paper should better discuss risks from synthetic cultural-value representation. LLM-generated stories may simplify or stereotype cultural, religious, political, or social values. The benchmark should not be used to infer real individuals’ private beliefs from ambiguous conversations.

**Claims And Evidence:**

No

**Claims Explanation:**

The experiments show that models differ on the proposed synthetic tasks, but they do not fully support the stronger claim that CQ-Bench measures realistic cultural intelligence. Since the stories are generated, revised, and validated mainly by GPT-family models, models may exploit synthetic artifacts rather than infer implicit values from natural human conversation. Human validation and the Reddit comparison are useful, but the latter is only a distinguishability test, not a labeled real-world benchmark subset.

The results could also be prompt/harness sensitive. The SFT gains should also be interpreted cautiously, since improvement from 500 synthetic examples may reflect benchmark-style adaptation rather than general cultural reasoning.

**Requested Changes:**

Critical:
1. Reframe CQ-Bench more clearly as a synthetic benchmark or synthetic data-generation framework, rather than a direct benchmark of real-world cultural intelligence.
2. Add a labeled real-conversation subset, for example from Reddit, and evaluate whether performance on synthetic CQ-Bench transfers to natural discourse.
3. Discuss the limitations of human validation more carefully, especially the moderate inter-annotator agreement and the relatively limited annotation scale.
4. Add a more systematic prompt/harness sensitivity analysis. At minimum, also report results under few-shot prompting.
5. Interpret the SFT/GRPO results more cautiously. The authors should test whether fine-tuned models generalize to stories generated by a different model family or to real conversations, to distinguish real cultural reasoning gains from benchmark-style adaptation.
6. Evaluate more model families, such as Claude.

Would strengthen the work:
1. Use multiple generator models, or compare with non-GPT-generated data, to reduce generator-specific bias.
2. Provide more examples and failure cases of generated stories.
3. Discuss whether all WVS/GlobalOpinion values are suitable for implicit conversational expression, since some survey-style values may be unnatural to embed in casual dialogue.

---

> ### Author Response · Authors · 2026-07-13
> **Questions regarding the real-conversation subset**
>
> Thank you for your constructive feedback! We are currently running additional experiments and generations for our paper. However, we would like to raise one issue regarding the real-conversation subset. The Reddit posts we sampled to reflect cultural values are individual posts rather than conversations. To address this, we propose synthesizing conversations that anchor multiple Reddit posts within a narrative. We generated 50 such conversations and conducted human validation, and the results appear reasonable. Before running more experiments, we would like to confirm whether this approach aligns with what you had in mind. Thank you!

---

> > ### Comment · Reviewer_gusa · 2026-07-13
> >
> > Thanks for asking. Would it be possible to use Reddit comment threads to identify a small set of real conversations? Even if the resulting subset is limited in scale or somewhat biased, I think it would still be a valuable complement to the synthetic dataset because the interactions would come directly from real users rather than being generated by an LLM.

---

> > > ### Author Response · Authors · 2026-07-13
> > >
> > > Thank you for your quick reply! We did in fact attempt this before submission, but found it challenging for several reasons. First, collecting utterances from Reddit was already highly resource-intensive: we ran extensive searches via the Reddit API across 200+ communities, followed by multiple filtering stages and LLM-assisted identification of relevant comments, which was quite costly. Second, since our goal is to construct coherent conversations spanning multiple topics with potentially differing attitudes, an LLM-based pre-filtering stage before human evaluation would require substantial additional time. We also tried manually browsing Reddit for suitable threads, but this proved inefficient.
> > > We will continue exploring ways to make this feasible. However, we would like to ask: if we are ultimately unable to include this in the final version, would this be a deal-breaker for you? Alternatively, would appropriately narrowing the scope of our claims better address your concern? We would greatly appreciate your guidance on which direction you would find most convincing.

---

> > > > ### Comment · Reviewer_gusa · 2026-07-13
> > > >
> > > > Would it be feasible to use publicly available Pushshift dumps to access a larger pool of Reddit discussions, and then apply rule-based filtering to reduce the candidate set before human review? I am not certain whether this would comply with Reddit’s current data-use policies, so please verify that carefully before proceeding.
> > > >
> > > > Regarding the impact on my evaluation, I could still lean toward borderline weak accept if the scope of the claims is clearly narrowed, with the synthetic nature of the benchmark emphasized in the title and consistently throughout the paper. However, including even a modest real-human subset—on the order of roughly 100 conversations, sourced from genuine interactions and human-validated—would make the work substantially more convincing and could move my recommendation toward accept.

---

> > > > > ### Author Response · Authors · 2026-07-13
> > > > >
> > > > > Thank you for the suggestion — pulling Reddit discussions is indeed something we can try. However, one inherent limitation is that each Reddit thread is anchored to a single stated topic, so the discussion tends to revolve around that one topic/value. This differs from natural human chitchat — for example, during a lunch break, people often shift topics frequently and unpredictably — which is precisely the conversational dynamic our dataset aims to capture.
> > > > > Given this constraint, here is what we propose: we can annotate real Reddit discussions as conversations reflecting a single value, but potentially with differing attitudes across participants. Accordingly, we would adapt the task formulation for this subset to (1) character-level attitude detection, and (2) multiple-choice (single-answer) value selection instead of multi-label selection. Would this design address your concern?

---

> > > > > > ### Comment · Reviewer_gusa · 2026-07-13
> > > > > >
> > > > > > I think this design would address the concern to some extent, although it would still be less ideal than identifying real discussions in which multiple topics arise naturally. If you can find such real-world conversations, or conduct a carefully controlled human study to collect them, that would be the strongest option—though I understand that the latter would be substantially more expensive, and I am not requiring it.
> > > > > >
> > > > > > If that is not feasible, a single-topic Reddit subset would still be a valuable addition. Human-annotating the conversations, cross-checking with an LLM, and using these annotations to cross-validate both the generation pipeline and the evaluation framework would meaningfully strengthen the paper.

---

### Author Response · Authors · 2026-07-16
**Request extension for review response**

Dear all,

We are writing to respectfully request a brief 5-day extension for submitting our response and revision.

The reviewers have provided highly constructive feedback, including suggestions for evaluations involving real human conversations and manual labeling. To address these thoroughly and rigorously, we require a bit more time, particularly as our current schedule directly overlaps with commitments for the ACL conference and the EMNLP rebuttal phase.

We kindly ask for this 5-day extension to ensure we can fully execute these additional evaluations and deliver a high-quality revision.

Thank you very much for your time, consideration, and understanding.

---

> ### Comment · Action_Editor_PeUQ · 2026-07-17
>
> No problem, go ahead. Please note that this may cut into reviewer response time though.